# Cluster Analysis to Distinguish Patients Most Likely to Benefit from Outpatient Cardiac Rehabilitation—A Prospective, Multicenter Study

**DOI:** 10.3390/ijerph191711000

**Published:** 2022-09-02

**Authors:** Jacek Hincz, Maciej Sterliński, Dariusz Kostrzewa, Rafał Dąbrowski, Edyta Smolis-Bąk

**Affiliations:** 1Medical Projects Department, COPERNICUS Healthcare Entity, 80-803 Gdansk, Poland; 2First Department of Cardiac Arrhythmias, National Institute of Cardiology, 04-628 Warsaw, Poland; 3COPERNICUS Healthcare Entity, 80-803 Gdansk, Poland; 4Department of Coronary Artery Disease and Cardiac Rehabilitation, National Institute of Cardiology, 04-628 Warsaw, Poland

**Keywords:** cardiac rehabilitation, cluster analysis, cholesterol, hypertension, uric acid, glucose

## Abstract

Offering cardiac rehabilitation to people who can benefit most could improve the outcomes in the context of limited availability. We used cluster analysis to distinguish three patient groups based on clinical and laboratory variables and then compared the outcomes of 6-month outpatient cardiac rehabilitation between these groups. The outcomes included blood pressure, blood lipids, fasting blood glucose, and uric acid concertation in serum. Group 1 consisted primarily of men with obesity, increased blood pressure, favourable lipid profiles and increased fasting glucose. Group 2 consisted of men or women with normal weight, normal blood pressure, favourable lipid profiles, and normal fasting glucose. Group 3 consisted primarily of women with overweight, normal blood pressure, unfavourable lipid profiles, and normal fasting glucose. After 6 months of cardiac rehabilitation, blood lipids improved in group 3, whereas blood pressure improved in groups 1 and 3, but the outcomes did not change significantly in group 2. We did not see any effect of cardiac rehabilitation on fasting blood glucose and serum uric acid concentration in any group. Concentrations of glucose and uric acid did not change significantly in any group. In conclusion, an adequate selection of patients should maximise the benefits of cardiac rehabilitation.

## 1. Introduction

Cardiac rehabilitation is a multifaceted intervention for patients with heart diseases. The pillars of cardiac rehabilitation include pharmacological management of the underlying disease and comorbidities, modification of lifestyle (healthy diet, exercise, non-smoking), and improvement of psychosocial well-being [1]. Therefore, cardiac rehabilitation requires cooperation between cardiologists, nurses, physical therapists, dietitians, and psychologists. Current evidence shows that cardiac rehabilitation can reduce mortality and hospital admission and improve the quality of life among patients with coronary heart disease, heart failure, or valvular disease [2,3,4,5].

More and more people need cardiac rehabilitation because of the increasing prevalence of cardiovascular diseases and the continuing improvements in cardiac care, which often enable many patients to survive acute events, such as acute coronary syndromes or decompensation of heart failure [6]. However, participation in cardiac rehabilitation is low around the world, which is largely because of limited access, particularly in low-income or middle-income countries [7,8]. Offering cardiac rehabilitation primarily to people who are most likely to benefit from it could improve the efficacy of this intervention from a public health standpoint.

Cluster analysis and other data-driven methods may help distinguish people who are most likely to benefit from health interventions and those who do not get any benefit [9,10]. Therefore, in this study, we used cluster analysis to distinguish different groups of patients eligible for outpatient cardiac rehabilitation based on several characteristics, and we compared outcomes between the groups. We aimed to find subgroups of patients who benefit most from physical therapy, psychological support, and dietary interventions and are therefore most likely to maintain their professional activity.

## 2. Materials and Methods

### 2.1. Study Setting and Rehabilitation Plan

The study was carried out in 10 outpatient centres of cardiac rehabilitation in the Pomeranian Voivodeship, Poland between January 2020 and May 2021 (see Appendix A for the full list of centres). The course of cardiac rehabilitation was planned for individual patients in compliance with the standards of the Polish Society of Cardiology [11]. At baseline, we carried out assessments to estimate exercise tolerance and risk of future cardiovascular events. The assessment included medical history, resting electrocardiography (ECG), an exercise ECG stress test, and 2D echocardiography. The course of rehabilitation consisted of 16–30 sessions lasting for up to 90 min each. Briefly, during the sessions, patients were instructed by cardiologists on the role of compliance with pharmacologic treatment, by physical therapists on how to exercise, by dieticians on the recommended diet, and by psychologists on the role to of lifestyle and how to change it. The sessions took place in outpatient clinics and at patients’ homes. Families of patients were educated on the role of compliance with treatment, exercise, and diet in reducing cardiovascular risk. Moreover, two and four months after the last session we contacted patients by phone to remind them about the recommendations.

### 2.2. Participants

Occupationally active patients aged 18–65 years with one or more of the following diseases: hypertension, ischemic heart disease, peripheral artery disease, cardiac arrhythmia, congenital heart defect, acquired heart defect, previous myocarditis, cardiac neoplasm, and previous ischemic stroke were eligible. The study was approved by the Ethics Committee by the Medical Chamber in Gdansk, Poland (KB, 22/21). All participants signed informed consent before enrolment.

### 2.3. Assessments

Data analysed in this study were gathered before the first and six months after the last rehabilitation course. The following data were recorded at baseline: sex, age, education (primary, occupational, secondary, tertiary), smoking (yes vs. no), social situation (living alone vs. with family), family history of cardiovascular diseases (yes vs. no), and medications for cardiovascular disease (yes vs. no). The following efficacy variables were measured at baseline and six months after the last rehabilitation course: body mass index (BMI), systolic blood pressure (SBP), diastolic blood pressure (DBP), pulse rate, total cholesterol (TC), low-density lipoprotein cholesterol (LDL-C), high-density lipoprotein cholesterol (HDL-C), non-HDL-C, triglycerides (TG), fasting glucose, and serum uric acid. Exercise capacity, in metabolic equivalents of task (METs), was measured at enrolment and study completion with the use of a cycle ergometer according to the Bruce protocol.

### 2.4. Statistical Analysis

Cluster analysis is an exploratory method in which objects (observations, patients) are allocated, based on object characteristics, to a set number of clusters. The clusters are distinguished by their centroids (most representative cluster points) and objects are allocated to the nearest centroid (smallest distance). Then, the values of centroids are modified, and objects are reallocated until they are no longer moving between clusters. We used cluster analysis based on k-means to distinguish three groups of patients based on baseline BMI, SBP, DBP, pulse rate, TC, LDL-C, HDL-C, non-HDL-C, TG, fasting glucose, and uric acid. For cluster analysis, all these variables were transformed into z-scores. The coefficient of determination (R2), Akaike Information Criterion (AIC), and Bayesian Information Criterion (BIC) were used to assess the performance of different models. The change in the efficacy variables for baseline to 6 months after the last rehabilitation course was compared between the groups with repeated-measures analysis of variance (ANOVA). An interaction term of group (groups yielded by cluster analysis) x time (baseline vs. 6 months after rehabilitation) was used to assess whether outcomes changed differently in particular groups. The JASP software (version 0.14.01, University of Amsterdam, Amsterdam, The Netherlands) was used for all analyses. A *p* < 0.05 was considered significant.

## 3. Results

### 3.1. Cluster Analysis and Baseline Characteristics

The initial result of cluster analysis based on baseline BMI, SBP, DBP, pulse rate, TC, LDL-C, HDL-C, non-HDL-C, TG, fasting glucose, and uric acid yielded three groups (R2 = 0.29; AIC = 5390.79; BIC = 5540.26). Serial ANOVAs showed significant differences between the three groups for all the variables except pulse rate (*p* = 0.440). Therefore, pulse rate was removed from cluster analysis, yielding an improved model with three groups (R2 = 0.32; AIC = 4766.67; BIC = 4902.90). Table 1 shows the means (standard deviations) for the variables used in cluster analysis by the group. Table 2 shows counts (percentages) for the remaining baseline variables.

Group 1 consisted mostly of men with secondary education, living with families, non-smoking, without a family history of cardiovascular disease, and receiving medications. The mean age of patients in group 1 was 54.94 ± 9.95 years. Group 2 consisted mostly of women with secondary or tertiary education, living with families, non-smoking, without a family history of cardiovascular diseases, and receiving medications. The mean age of patients in group 2 was 53.01 ± 7.94 years. Group 3 consisted mostly of women, with tertiary education, living with families, non-smoking, without a family history of cardiovascular diseases, and receiving medications. The mean age of patients in group 3 was 49.34 ± 11.06 years.

The mean BMI was normal in group 2 only, whereas the mean BMI in group 3 indicated overweight and that in group 1 indicated obesity (Table 2). The mean SBP was increased above 140 mm Hg in group 1 only, whereas the mean DBP was increased above 80 mmHg in groups 1 and 3. The mean LDL-C was substantially increased in group 3, whereas it was below 100 mg/dL in groups 1 and 2. The mean TG concentration was lower in group 2 than in groups 1 and 3. Similarly, the mean concentration of uric acid was somewhat lower in group 2 than in groups 1 and 3, but it was within the normal range in all groups. The mean fasting glucose concentration was increased in group 1 only.

Overall, group 1 could be characterised as primarily men with secondary education, obesity, increased blood pressure, favourable lipid profiles and increased fasting glucose. Group 2 could be characterised as men or women with secondary or tertiary education, normal weight, normal blood pressure, favourable lipid profiles, and normal fasting glucose. Group 3 could be characterised as primarily women, with tertiary education, overweight, normal blood pressure, unfavourable lipid profiles, and normal fasting glucose.

### 3.2. Change in Outcomes from Baseline to 6 Months after the End of Cardiac Rehabilitation

Repeated measures ANOVA showed that SBP and DBP decreased significantly from baseline to 6 months after cardiac rehabilitation in group 1 only [interaction term for SBP: F(2, 690) = 9.03, *p* < 0.001; interaction term for DBP: F(2, 690) = 7,20, *p* < 0.001; Figure 1]. Figure 1 shows point estimates and 95% confidence intervals for all groups at baseline and 6 months after cardiac rehabilitation.

TC decreased significantly in group 3 (*p* < 0.001 for interaction term), whereas it increased significantly in group 2 [interaction term: F(2, 690) = 31.08; *p* < 0.001, Figure 2A]. LDL-C decreased significantly in group 3 only [interaction term: F(2, 690) = 34.96; *p* < 0.001, Figure 2B]. HDL-C increased significantly in all groups [F(1, 690) = 14.05; *p* < 0.001], with a non-significant group x time interaction [F(2, 690) = 0.21; *p* = 0.815; Figure 2C]. Non-HDL cholesterol decreased significantly in group 3, whereas it increased significantly in group 2 [interaction term: F(2, 690) = 42.70, *p* < 0.001; Figure 2D]. Although TG changed differently in the three groups (interaction term: F(2, 690) = 6.75, *p* = 0.001], overall, the change from baseline to 6 months after cardiac rehabilitation was non-significant (Figure 2E). Fasting glucose did not change significantly from baseline to 6 months after cardiac rehabilitation in the three groups [main effect of time: F(1, 690) = 1.87, *p* = 0.171; interaction term: F(1, 690) = 1.07, *p* = 0.342; Figure 2F], and neither did uric acid [main effect of time: F(1, 680) = 1.33; *p* = 0.249; interaction term: F(1, 680) = 0.09, *p* = 0.918; Figure 2G]. The mean pulse rate did not change significantly in any of the groups [main effect of time: F(1, 690) = 0.13, *p* = 0.718; interaction term: F(1, 690) = 1.94, *p* = 0.144; data not shown).

The improvement in exercise capacity (METs) at the end of the rehabilitation course differed significantly between the three groups [F(2, 688) = 3.26; *p* = 0.039], with groups 1 and 3 showing a measurable improvement (see Figure 3). On post hoc testing, only the difference between groups 1 and 2 remained significant (*p* = 0.035). The BMI did not change substantially at the end of the rehabilitation course, with the mean changes not differing significantly between the three groups [F(2, 690) = 1.31; *p* = 0.323)].

## 4. Discussion

This study, using a data-driven approach, showed that the benefits of cardiac rehabilitation are greater in certain groups of patients, whereas in others cardiac rehabilitation may provide no significant benefit. Of the three groups distinguished by cluster analysis, group 3 consisted mainly of women with overweight, normal BP, unfavourable lipid profiles, and normal blood glucose. In this group, blood lipids showed consistent improvement. Blood pressure improved in groups 1 and 3. Improvement in exercise tolerance was measurable in groups 2 and 3. We did not see any effect of cardiac rehabilitation on fasting blood glucose and serum uric acid concentrations in any of the groups. 

Cardiac rehabilitation is defined by the WHO as “the sum of activities required to influence favourably the underlying cause of the disease, as well as the best possible physical, mental and social conditions, so that they may, by their own efforts preserve or resume when lost, as normal a place as possible in the community.” [12]. This general definition illustrates that there is no universal way to provide cardiac rehabilitation to all patients and therefore cardiac rehabilitation must be individualised. Cardiac rehabilitation is an effective intervention that helps control the most important modifiable cardiovascular risk factors.

Previous studies showed that cardiac rehabilitation significantly reduces SBP and DBP. Among 35 patients after myocardial infarction, Kargarfard et al. showed that aerobic training over 2 months significantly reduced both SBP and DBP [13]. Similarly, among 30 men with stable angina, SBP and DBP decreased significantly after six weeks of aerobic exercise performed four days per week [14]. Importantly, Parvand et al. showed that a reduction in SBP in patients after myocardial infarction who took part in a cardiac rehabilitation preprogramme was associated with improved functional capacity [15]. Mohammed and Shabana reported that cardiac rehabilitation significantly reduced blood pressure and heart rate in patients with chronic heart failure [16]. SBP was also reduced following cardiac rehabilitation in a retrospective, single-centre study among patients requiring primary or secondary prevention [17]. Similarly, Sahn et al. reported that blood pressure decreased in hypertensive patients who took part in a cardiac rehabilitation program but not in those who refused to participate [18]. In a registry-based study from Austria, outpatient cardiac rehabilitation improved exercise capacity, blood pressure, blood lipids, and glucose [19]. In our study, SBP and DBP decreased in group 1 (significantly) and group 3 (non-significantly) and these groups had a measurable improvement in exercise capacity. In these two groups, baseline SBP and DBP values were considerably higher than in group 2, in which the average values at baseline were within the normal range. Similarly, patients in group 1 and group 3 were on average obese or overweight, whereas those in group 2 had normal weight. Likely because blood pressure was well controlled before the start of cardiac rehabilitation, the mean SBP and DBP values did not change in group 2. In contrast, in groups 1 and 3, the interventions used in our study, including the adjustment of medications and exercises, reduced SBP and DBP.

Cardiac rehabilitation can improve blood lipids. Adeba-Garcia et al. showed that LDL-C decreased significantly 6 months after an acute coronary syndrome among 268 prospectively followed patients who received a multi-faceted cardiac rehabilitation [20]. Lavie and Milani reported that LDL-C and TG decreased, whereas HDL-C increased after cardiac rehabilitation consisting of an exercise program in over 300 patients with coronary artery disease and baseline hypertriglyceridemia [21]. Similarly, in the study by Fard et al., a cardiac rehabilitation programme with 24 exercise sessions and dietary and psychiatric consultations significantly reduced LDL-C and increased HDL-C [22]. Among nearly 1500 patients after myocardial infarction, Shwaab et al. reported that the proportion of patients achieving LDL-C < 70 mg/dL (1.8 mmol/L) or at least 50% reduction of baseline value increased from ~2% at baseline to ~42% at follow-up after a mean of 22 days of comprehensive cardiac rehabilitation [23]. Among patients with ischemic heart disease, LDL-C concentrations decreased after 3 months of cardiac rehabilitation and remained reduced for up to 12 months [24]. In patients after myocardial infarction, high-intensity but not moderate-intensity exercise significantly reduced LDL-C and TG [25]. Similarly, in another study among patients with various cardiovascular diseases, only intensive but not standard cardiac rehabilitation reduced LDL-C and other atherogenic blood lipids [26]. In that study, both standard and intensive rehabilitation improved exercise tolerance [26], which is in line with our study. In our study, LDL-C decreased in group 3 only, in which the baseline LDL-C values were substantially greater than in the remaining groups. Although cardiac rehabilitation was associated with a reduction in TG in previous reports [27], TG concentration did not change significantly in our study. Similarly, although cardiac rehabilitation can improve glycaemic control in patients with diabetes [28], we did not see any change in fasting plasma glucose. Likewise, uric acid concertation, which is associated with the risk of cardiovascular diseases, did not change after cardiac rehabilitation in our study [29].

The availability of cardiac rehabilitation is very low globally, with only ~40% of countries offering cardiac rehabilitation programmes [30]. In Europe, less than half of eligible patients receive cardiac rehabilitation, which is partly because of inadequate provision of these services [31]. Thus, selecting patients most likely to benefit from cardiac rehabilitation might optimise the effects of this intervention from a public health standpoint. In our study, group 2, which was characterised by normal values of the cardiovascular risk factors, did not benefit substantially from cardiac rehabilitation. However, such patients should still undergo cardiac rehabilitation to obtain long-term benefits, such as reduced mortality. When resources are limited though, it seems that patients with the greatest cardiovascular burden would see the greatest improvement.

Our study had limitations. The group of patients was heterogenous because we included patients with different cardiovascular diseases. Moreover, the use of a data-driven approach makes it difficult to interpret the results in clinical terms. The generalisability of our results is limited because of a substantial proportion of women in our sample, whereas secondary prevention of cardiovascular diseases is more common in men [32].

## 5. Conclusions

In conclusion, developing criteria for an adequate selection of patients should maximise the benefits of cardiac rehabilitation in the context of limited availability.

## Figures and Tables

**Figure 1 ijerph-19-11000-f001:**
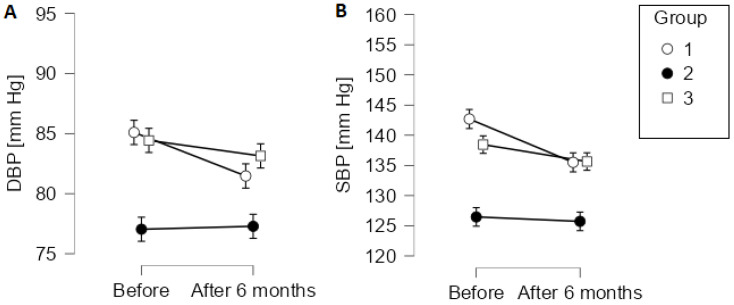
Change in diastolic (**A**) and systolic blood pressure (**B**) from baseline to 6 months after cardiac rehabilitation by group. Error bars show 95% confidence intervals. DBP, diastolic blood pressure; SBP, systolic blood pressure.

**Figure 2 ijerph-19-11000-f002:**
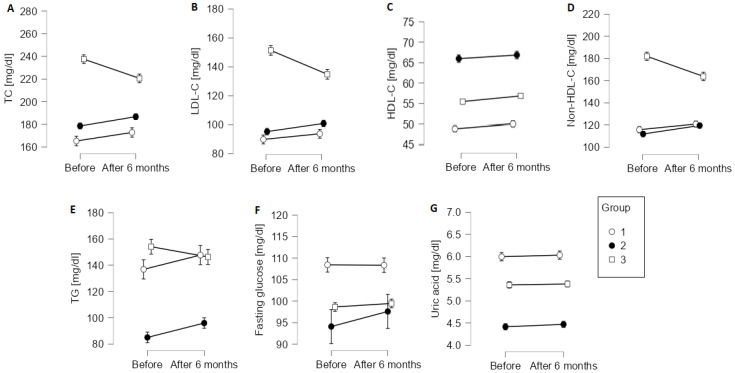
Change in laboratory variables from baseline to 6 months after completion of a course of cardiac rehabilitation in patient groups distinguished by cluster analysis. Error bars show 95% confidence intervals. TC, total cholesterol (**A**); LDL-C, low-density lipoprotein cholesterol (**B**); HDL-C, high-density lipoprotein cholesterol (**C**); Non-HDL-C, non high-density lipoprotein cholesterol (**D**); TG, triglycerides (**E**); Fasting glucose (**F**); Uric acid (**G**).

**Figure 3 ijerph-19-11000-f003:**
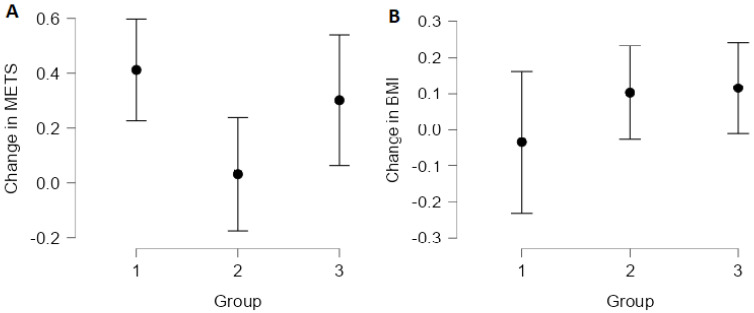
Change in exercise capacity (METS, (**A**)) and BMI (**B**) in the three study groups.

**Table 1 ijerph-19-11000-t001:** Baseline characteristics (continuous variables) for the three groups yielded by cluster analysis.

	Group 1	Group 2	Group 3
**BMI**	31.06 (4.52)	24.45 (3.34)	28.88 (4.00)
**SBP (mm Hg)**	142.69 (16.46)	126.46 (14.42)	138.45 (16.25)
**DBP (mm Hg)**	85.09 (10.40)	77.04 (8.67)	84.43 (9.96)
**Pulse rate (/min)**	75.15 (12.31)	73.87 (14.00)	74.38 (12.83)
**TC (mg/dL)**	165.39 (27.18)	178.59 (32.11)	237.54 (26.79)
**LDL-C (mg/dL)**	89.62 (23.16)	95.38 (26.08)	151.23 (24.44)
**HDL-C (mg/dL)**	48.91 (11.65)	66.01 (14.36)	55.51 (12.50)
**Non-HDL-C (mg/dL)**	115.80 (26.23)	111.89 (26.90)	181.96 (25.83)
**TG (mg/dL)**	137.07 (61.35)	85.00 (31.85)	154.12 (61.70)
**Fasting glucose (mg/dL)**	108.40 (18.02)	94.13 (9.34)	98.65 (12.94)
**Uric acid (mg/dL)**	5.98 (1.13)	4.42 (0.99)	5.35 (1.03)
**Age years**	54.94 (9.95)	53.01 (7.94)	49.34 (11.06)

BMI, body mass index; SBP, systolic blood pressure; DBP, diastolic blood pressure; TC, total cholesterol; LDL-C, low-density lipoprotein cholesterol; HDL-C, high-density lipoprotein cholesterol; TG, triglycerides. Values are means (standard deviations).

**Table 2 ijerph-19-11000-t002:** Baseline characteristics (categorical variables) for the three groups yielded by cluster analysis.

		Group 1		Group 2		Group 3	
		*n*	%	*n*	%	*n*	%
**Sex**	Women	75	32.33	139	58.65	159	70.98
	Men	157	67.67	98	41.35	65	29.02
**Education**	Primary	2	0.86	3	1.27	4	1.79
	Occupational	34	14.66	20	8.44	20	8.93
	Secondary	118	50.86	105	44.30	81	36.16
	Tertiary	78	33.62	109	45.99	119	53.13
**Smoking**	No	210	90.52	202	85.23	196	87.50
	Yes	22	9.48	35	14.77	28	12.50
**Social situation**	Lives alone	11	4.74	12	5.06	18	8.04
	Lives with family	221	95.26	225	94.94	206	91.96
**Family history**	No	162	69.83	153	64.56	165	73.66
	Yes	70	30.17	84	35.44	59	26.34
**Medications**	No	25	10.82	62	26.61	88	39.46
	Yes	206	89.18	171	73.39	135	60.54

## Data Availability

Data are available from the corresponding author.

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
