# Peer review of "Cluster Analysis to Distinguish Patients Most Likely to Benefit from Outpatient Cardiac Rehabilitation—A Prospective, Multicenter Study"

_ijerph, 2022, doi:10.3390/ijerph191711000_

Round 1
Reviewer 1 Report
Dear Authors
That is an interesting study focusing on additional effects of rehabilitation program on classical risk factor reduction in clustered groups of patients.
The aim of the study is to describe some profiles of patients which can maximally benefit of rehabilitation program which is for me very interesting and worth studing
My main critical remarks are:
- One of he main goals of rehabilitation programs is an increase “in general fitness “ and exercise capacity. In the factors selected for clustering or even in discussion this goal was not mentioned. It is not clear for me whether this effect expressed in exercise capacity or other measures of fitness (except heart rate and blood pressure) was similar in all clusters
- It is not clear for me what was the adherence to the program – how many patients were included initially and how many concluded the program and were included in the analyses. Was it similar in all clusters?
- In my opinion the effect on overall increase of fitness by measurable effect (if done in the study eg, exercise tests in watts, 6 minutes walk test, exercise capacity in stress test) should be mentioned –especially whether it was similar or different in all clusters mentioned in this article or changed as an effect of rehabilitation program
- To illustrate this please add an additional chart similar to those in Fig. 2 with some other measurable fitness results initially and after 6 months
- Readers not involved directly in rehabilitation programs could interpret those results as that patients in some cluster do not benefit in any way from rehabilitation and should not be rehabilitated- I do not think this was the idea of the Authors. Please clearly state this in the discussion section.
Author Response
Dear Sir/Madam,
We wish to thank you for the critical comments on our manuscript titled “Cluster analysis to distinguish patients most likely to benefit from outpatient cardiac rehabilitation - a prospective, multi-center study”. Below, we respond to the comments point-by-point.
That is an interesting study focusing on additional effects of rehabilitation program on classical risk factor reduction in clustered groups of patients. The aim of the study is to describe some profiles of patients which can maximally benefit of rehabilitation program which is for me very interesting and worth studying
Thank you for this encouraging remark.
One of he main goals of rehabilitation programs is an increase “in general fitness “ and exercise capacity. In the factors selected for clustering or even in discussion this goal was not mentioned. It is not clear for me whether this effect expressed in exercise capacity or other measures of fitness (except heart rate and blood pressure) was similar in all clusters.
Thank you or suggesting further analyses. We measured exercise capacity at baseline and at the end of cardiac rehabilitation with a cycle ergometer. We added these data in the manuscript.
In the methods: “Exercise capacity, in metabolic equivalents of task (METs), was measured at enrolment and study completion with the use of a cycle ergometer according to the Bruce protocol.”
In the results: “The improvement in exercise capacity (METs) at the end of the rehabilitation course differed significantly between the three groups [F(2, 688) = 3,26; p = 0.039], with only groups 1 and 3 showing a measurable improvement (see Fig. 3). On post-hoc testing, only the difference between group 1 and 2 remained significant (p = 0.035).”
In the discussion: “Improvement in exercise tolerance was measurable in groups 2 and 3”.
It is not clear for me what was the adherence to the program – how many patients were included initially and how many concluded the program and were included in the analyses. Was it similar in all clusters?
Only patients who completed the program as planned were included in the analysis. Any patient could opt out at any time without giving any reason. 437 patients (38.67%) did not complete the program. As the project was addressed to professionally active and actively working patients, professional commitments, lack of time and generally poor compliance were usually the causes of premature termination or termination of the program. Unfortunately, we do not have detailed information or a list of patients or reasons for the cancellation. The resignation took place only at the patient's request. There have been no reports that the program had to be interrupted or shortened due to medical indications or as a result of an adverse medical event.
In my opinion the effect on overall increase of fitness by measurable effect (if done in the study eg, exercise tests in watts, 6 minutes walk test, exercise capacity in stress test) should be mentioned –especially whether it was similar or different in all clusters mentioned in this article or changed as an effect of rehabilitation program. To illustrate this please add an additional chart similar to those in Fig. 2 with some other measurable fitness results initially and after 6 months.
We added further analyses on exercise capacity (please see the previous comment). Additionally, we present data on the change in BMI at the end of the study. We added the following sentence in the results section: “The BMI did not change substantially at the end of the rehabilitation course, with the mean changes not differing significantly between the three groups [F(2, 690) = 1,31; p = 0.323)].”
Readers not involved directly in rehabilitation programs could interpret those results as that patients in some cluster do not benefit in any way from rehabilitation and should not be rehabilitated- I do not think this was the idea of the Authors. Please clearly state this in the discussion section.
Of course, we agree. To make it clear, we added the following sentence in the discussion: “However, such patients should still undergo cardiac rehabilitation to obtain long-term benefits, such as reduced mortality. When resources are limited though, it seems that patients with the greatest cardiovascular burden would see the greatest improvement”.
Sincerely,
Jacek Hincz

Reviewer 2 Report
No analysis is made on the evolution of the BMI in the group of overweight patients and in the group of obese patients.
In group two there aren´t changes in the risk factors 6 months after completing cardiac rehabilitation, because there aren´t cardiovascular risk factors in this group.
The cardiovascular pathologies of the sample are not mentioned. The large number of women is striking, since there are usually more men in the samples of patients with ischemic heart disease.
Some of the cited references are not recent publications (within the last 5 years) and there are citations that aren´t relevant.
Author Response
Dear Sir/Madam,
We wish to thank you for the critical comments on our manuscript titled “Cluster analysis to distinguish patients most likely to benefit from outpatient cardiac rehabilitation - a prospective, multi-center study”. Below, we respond to the comments point-by-point.
No analysis is made on the evolution of the BMI in the group of overweight patients and in the group of obese patients.
We agree that analyzing BMI is advantages. We added the following sentence: “The BMI did not change substantially at the end of the rehabilitation course, with the mean changes not differing significantly between the three groups [F(2, 690) = 1,31; p = 0.323)].”
In group two there aren´t changes in the risk factors 6 months after completing cardiac rehabilitation, because there aren´t cardiovascular risk factors in this group.
We agree with this remark. In the discussion, we write: “group 2, which was characterized by normal values of the cardiovascular risk factors, did not benefit substantially from cardiac rehabilitation.”. This observation shows that appropriate patient selection for cardiac rehabilitation is crucial when rehabilitation resources are limited, as some people may not see measurable effects.
The cardiovascular pathologies of the sample are not mentioned.
In the methods section, we now provide detailed eligibility criteria: “Eligible were occupationally active patients aged 18-65 years with one or more of the following diseases: hypertension, ischemic hear disease, peripheral artery disease, cardiac arrhythmia, congenital heart defect, acquired heart defect, previous myocarditis, cardiac neoplasm, and previous ischemic stroke.” Unfortunately, we did not gather data on these comorbidities.
The large number of women is striking, since there are usually more men in the samples of patients with ischemic heart disease.
Indeed. Cardiovascular disease is more common in men than women. In the limitations, we added the following sentence: “The generalizability of our results is limited because of a substantial proportion of women in our sample, whereas secondary prevention of cardiovascular diseases is more common in men [1]”.
Some of the cited references are not recent publications (within the last 5 years) and there are citations that aren´t relevant.
The studies that have analyzed the effects of cardiac rehabilitation on cardiovascular risk factors measured in our study (e.g., blood pressure, lipids) are scarce. Instead, most studies have investigated outcomes such as the rates of death or hospitalization, which our stud did not assess. Therefore, we refer to all available studies with results that could be related to ours. We agree, however, that adding newer evidence is important. Therefore, after a thorough literature search, we added new relevant references in the discussion:
“Mohammed and Shabana reported that cardiac rehabilitation significantly reduced blood pressure and heart rate in patients with chronic heart failure [16]. SBP was also reduced following cardiac rehabilitation in a retrospective, single-center study among patients requiring primary or secondary prevention [17]. Similarly, Sahn et al. reported that blood pressure decreased in hypertensive patients who took part in a cardiac rehabilitation program but not in those who refused to participate [18]. In a registry-based study from Austria, outpatient cardiac rehabilitation improved exercise capacity, blood pressure, blood lipids, and glucose [19].”
“Among patients with ischemic heart disease, LDL-C concentrations decreased after 3 months of cardiac rehabilitation and remained reduced for up to 12 months [24]. In patients after myocardial infarction, high-intensity but not moderate-intensity exercise significantly reduced LDL-C and TG [25]. Similarly, in another study among patients with various cardiovascular diseases, only intensive but not standard cardiac rehabilitation reduced LDL-C and other atherogenic blood lipids [26]. In that study, both standard and intensive rehabilitation improved exercise tolerance [26], which is in line with our study.
We hope that you will find the improved manuscript suitable for publication in the esteemed “International Journal of Environmental Research and Public Health”. Of course, we would be glad to provide further information.
Sincerely,
Jacek Hincz

Round 2
Reviewer 2 Report
Thanks for the clarifications, but we still need to explain the percentage of patients with each pathology
Author Response
Dear Sir/Madam,
We wish to thank you for the critical comments on our manuscript titled “Cluster analysis to distinguish patients most likely to benefit from outpatient cardiac rehabilitation - a prospective, multi-center study”. Below, we respond to the comments point-by-point.
Thanks for the clarifications, but we still need to explain the percentage of patients with each pathology.
Unfortunately, we do not have data of percentage of patients with each pathology. This program was aimed at the general population with cardiovascular disease and was not intended to assess morbidity or incidence.
We hope that you will find the improved manuscript suitable for publication in the esteemed “International Journal of Environmental Research and Public Health”. Of course, we would be glad to provide further information.
Sincerely,
Jacek Hincz